# Indirect Treatment Comparison of First-Line CDK4/6-Inhibitors in Post-Menopausal Patients with HR+/HER2− Metastatic Breast Cancer

**DOI:** 10.3390/cancers15184558

**Published:** 2023-09-14

**Authors:** Joseph J. Zhao, Khi Yung Fong, Yiong Huak Chan, Jeremy Tey, Shaheenah Dawood, Soo Chin Lee, Richard S. Finn, Raghav Sundar, Joline S. J. Lim

**Affiliations:** 1Yong Loo Lin School of Medicine, National University of Singapore, 1E Kent Ridge Road, Singapore 119228, Singapore; jzhaozw@hotmail.com (J.J.Z.); khiyung@gmail.com (K.Y.F.); csilsc@nus.edu.sg (S.C.L.); 2Department of Haematology-Oncology, National University Cancer Institute, National University Hospital, Singapore 119074, Singapore; 3Biostatistics Unit, Yong Loo Lin School of Medicine, National University of Singapore, Singapore 119228, Singapore; medcyh@nus.edu.sg; 4Department of Radiation Oncology, National University Cancer Institute, National University Hospital, Singapore 119074, Singapore; jeremy_tey@nuhs.edu.sg; 5Department of Medical Oncology, Mediclinic City Hospital, Mohammed Bin Rashid University of Medicine and Health Sciences, Dubai 505055, United Arab Emirates; shaheenah@post.harvard.edu; 6Cancer Science Institute, National University of Singapore, Singapore 119074, Singapore; 7Department of Medicine, Division of Hematology and Oncology, David Geffen School of Medicine, University of California, Los Angeles, CA 90024, USA; rfinn@mednet.ucla.edu; 8Cancer and Stem Cell Biology Program, Duke-NUS Medical School, Singapore 169547, Singapore; 9The N.1 Institute for Health, National University of Singapore, Singapore 117456, Singapore; 10Singapore Gastric Cancer Consortium, Singapore 119074, Singapore

**Keywords:** metastatic breast cancer, CDK4/6-inhibitor

## Abstract

**Simple Summary:**

CDK4/6-inhibitors are an effective first-line treatment for patients with HR+/HER2− metastatic breast cancer (MBC). We aimed to compare overall survival (OS) and progression-free survival (PFS) between three CDK4/6-inhibitors from randomized controlled trials via a graphical reconstructive algorithm. No significant OS and PFS differences were found between palbociclib, ribociclib, and abemaciclib, supporting all three drugs as feasible options in first-line treatment in combination with endocrine therapy for post-menopausal patients with metastatic HR+/HER2− MBC.

**Abstract:**

**Background:** CDK4/6-inhibitors have demonstrated similar efficacy and are considered an effective first-line endocrine treatment of patients with hormone-receptor positive (HR+)/human-epidermal-growth-factor-receptor-2 negative (HER2−) metastatic breast cancer (MBC) in the endpoint of progression-free survival (PFS). Amongst these, palbociclib was first to achieve regulatory approval, followed subsequently by ribociclib and abemaciclib. However, recent updates of overall survival (OS) showed inconsistencies in the OS benefit for palbociclib compared with the other two CDK4/6-inhibitors. With the lack of head-to-head comparison studies, our study sought to compare indirect survival outcomes between CDK4/6-inhibitors in this setting using a novel reconstructive algorithm. **Methods:** Phase III randomized trials comparing first-line aromatase inhibitor with/without a CDK4/6-inhibitor in post-menopausal patients with HR+/HER2− MBC were identified through systemic review and literature search of online archives of published manuscripts and conference proceedings. A graphical reconstructive algorithm was utilized to retrieve time-to-event data from reported Kaplan-Meier OS and PFS plots to allow for comparison of survival outcomes. Survival analyses were conducted with Cox proportional-hazards model with a shared-frailty term. **Results:** Three randomized phase III trials—PALOMA-2, MONALEESA-2 and MONARCH-3—comprising 1827 patients were included. Indirect pairwise comparisons of all CDK4/6-inhibitors showed no significant PFS differences (all *p* > 0.05). Likewise, indirect treatment comparison between ribociclib vs. palbociclib (one-stage: HR = 0.903, 95%-CI: 0.746–1.094, *p* = 0.297), abemaciclib vs. palbociclib (one-stage: HR = 0.843, 95%-CI: 0.690–1.030, *p* = 0.094) and abemaciclib vs. ribociclib (one-stage: HR = 0.933, 95%-CI: 0.753–1.157, *p* = 0.528) failed to demonstrate a significant OS difference. **Conclusions:** Findings from this indirect treatment comparison suggest no significant PFS or OS differences between CDK4/6-inhibitors in post-menopausal patients with HR+/HER2− MBC.

## 1. Introduction

The introduction of cyclin-D–cyclin-dependent kinase 4/6-inhibitors (CDK4/6-inhibitor) over the past decade has led to significant improvement in the landscape of patients with hormone receptor-positive (HR+)/human epidermal growth factor receptor 2 negative (HER2−) metastatic breast cancer (MBC). Landmark randomized controlled trials (RCTs) evaluating this combination in post-menopausal women (PALOMA-2 [1], MONALEESA-2 [2] and MONARCH-3 [3]) have demonstrated significant improvements in progression-free survival (PFS), establishing this combination as the standard of care in first line endocrine treatment of patients with HR+/HER2− MBC.

In light of similar outcomes across the landmark RCTs for the primary endpoint of PFS, the drug efficacy of CDK4/6-inhibitors palbociclib, ribociclib and abemaciclib have been deemed comparable, and the eventual choice of agent in a clinical setting is largely led by cost, toxicity profile, and physician preference [4]. Nonetheless, recent updates of overall survival (OS) data has showed discrepancies in OS benefit, leading to concerns of comparability between the three CDK4/6-inhibitors. While MONALESSA-2 [5] has demonstrated a significant overall survival (OS) benefit with the addition of the CDK4/6-inhibitor ribociclib to letrozole and MONARCH-3 [6] showed a strong trend towards OS significance with the addition of abemaciclib to endocrine therapy in its interim analysis, PALOMA-2 [7] did not demonstrate similar statistical improvement in OS with the addition of palbociclib to letrozole. The seemingly inconsistent results in terms of OS seen in PALOMA-2 have been attributed to a myriad of reasons—including a large and disproportionate amount of missing data resulting from high dropout rates and loss to follow-up in both arms of the study, the subsequent exposure to CDK4/6-inhibitor agents in the control arm at disease progression, a patient population with poorer prognosis in view of the inclusion of 20% of patients that recurred while on or within 12 months of completing adjuvant therapy, and true potential efficacy differences in terms of effects on OS between palbociclib and the other two CDK 4/6 inhibitors.

In clinical practice, the discrepancy in OS benefit across these three landmark studies despite remarkably similar hazard ratios in the primary endpoint of PFS challenges our conventional notions of comparability, putting forth a conundrum in treatment choice when faced with a patient. With the lack of head-to-head RCTs comparing the three agents, we sought to compare survival outcomes between three different CDK4/6-inhibitor agents to provide additional evidence towards treatment choices. Advances in graphical plot digitization and computational inference now allows for derivation of individual patient data directly from graphs and figures presented in manuscripts and conference abstracts. This technique is now gaining traction allowing for large individual patient-data secondary analyses to be performed [8,9,10,11]. Hence, we aimed to harness this technique to conduct a patient-level indirect treatment comparison of OS and PFS between first-line CDK4/6-inhibitor agents in HR+/HER2− metastatic breast cancer.

## 2. Methodology

### 2.1. Search Strategy and Study Selection

This study was conducted in accordance with the Preferred Reporting Items for Systematic Reviews and Meta-Analyses (PRISMA) guidelines [12]. The search string utilized was detailed in Appendix A. Articles reporting phase III prospective randomized controlled trials comparing first-line aromatase inhibitor with or without a CDK4/6-inhibitor in post-menopausal patients with HR+/HER2− MBC were included in this analysis. Both original trial publications and conference abstracts were included. A hand search was performed for additional articles through the reference lists of obtained articles and conference proceedings from the American Society of Clinical Oncology Annual Meeting (ASCO) and European Society for Medical Oncology Congress (ESMO) up to 2022. This search was conducted without language restriction. Two authors (K.Y.F., J.J.Z.) independently filtered the title abstracts, followed by full-text review. Discrepancies were resolved by consensus or in consultation with a senior author (J.S.J.L.). 

### 2.2. Quality Assessment

The methodologic quality of the included studies for the primary and secondary outcomes were assessed using the revised Cochrane risk-of-bias tool for randomized trials (RoB2) which scores the risk of bias in five domains namely: randomization process, deviations from the intended interventions, missing outcome data, measurement of the outcome and selection of the reported result. The Cochrane Risk-of-Bias assessment was utilized for quality assessment of included articles. The Cochrane RoB assessment was done independently by two authors (K.Y.F., J.J.Z.) and any discrepancies were resolved by a senior author (J.S.J.L.). Quality of evidence was evaluated using the Grading of Recommendations, Assessment, Development and Evaluations (GRADE) framework. The domains assessed included risk of bias, inconsistency, indirectness, imprecision, publication bias, magnitude of effect, plausible confounding and dose-response relationship.

### 2.3. Data Extraction

Data was extracted using a standardized form. Extracted data included patient characteristics (number of patients, age, ECOG, disease-free interval [DFI]), tumor characteristics (hormone receptor status, prior chemotherapy, prior endocrine therapy, number of metastatic sites and visceral organ involvement) and clinical outcomes (OS and PFS). The primary outcome of interest was OS. Secondary outcome of interest was PFS. 

A graphical reconstructive algorithm was exploited to estimate time-to-event outcomes from reported Kaplan-Meier (KM) plots by methods described by Guyot et al. [13] Briefly, images of KM plots were extracted from original reports and digitized. The graphical reconstructive algorithm was employed to reconstruct patient-level survival data from digitized KM curves by back-solving the KM equations, utilizing data from the risk table [13]. Hazard ratios (HRs) were computed from these individual patient data using a marginal Cox proportional hazards model. Reconstructed KM plots were compared to original plots by inspecting the visual shape of plots, marginal HRs, log-rank values, median overall survival (OS)/progression free survival (PFS) time and number-at-risk tables. Where multiple PFS plots were provided, response assessment derived by blinded, independent central review was utilized.

### 2.4. Statistical Analysis

Subsequently, the retrieved time-to-event data was pooled together with shared-frailty term incorporated to account for between-study differences in a Cox proportional hazards model—thereby assuming that patients within each study are similarly failure-prone as others belonging to the same study. Comparisons between CDK4/6-inhibitor agents were subsequently derived. The proportional hazards assumption was evaluated with plots of scaled Schoenfeld residuals. As a sensitivity analysis, conventional two-stage indirect treatment comparison with study-level HRs was conducted. 

All analyses were conducted in R-4.1.0 with packages IPDfromKM [14], netmeta and survival and a two-sided *p*-value < 0.05 was regarded to indicate statistical significance.

## 3. Results

### 3.1. Search Strategy, Outcomes of Data Extraction

A search was conducted from inception to 15 September 2022 across PubMed, Embase, American Society of Clinical Oncology Annual Meeting (ASCO) 2022, European Society for Medical Oncology Congress (ESMO) 2022. The search strategy yielded 842 records from PubMed and Embase and 2 records from ASCO 2022 and ESMO 2022 after de-duplication (Figure 1). After screening and assessment for eligibility, 6 reports from 3 trials PALOMA-2 [1,7], MONALEESA-2 [2,5] and MONARCH-3 [3,6] were included in the analysis. The graphical reconstructive algorithm of OS and PFS KM plots yielded patient-level data that derived similar HRs and log-rank values to original plots (Appendix A). 

### 3.2. Baseline Characteristics of Trials

A summary of baseline trial characteristics may be found in Table 1. Of note, patients in PALOMA-2 were noted to have a greater proportion of patients with a less than 12-month treatment DFI (palbociclib arm—22.3%; placebo arm—21.6%) compared to MONARCH-3 (patients with disease-free interval <12 months were excluded) and MONALEESA-2 (ribociclib arm—1.2%; placebo arm—3.0%) (Table 1). A greater proportion of patients enrolled in MONARCH-3 had newly diagnosed metastatic disease (abemaciclib arm—41.2%, placebo arm—37.0%) (Table 1). 

### 3.3. Assessment of Network, Inconsistency, and the Transitivity Assumption

Statistical consistency was not evaluable in view of the open nature of networks (Figure 2) [15]. The transitivity assumption was evaluated clinically [16].

### 3.4. Progression-Free Survival

Three phase III randomized controlled trials, PALOMA-2 [1], MONALEESA-2 [2] and MONARCH-3 [3], comprising 1827 patients were included in this analysis. Indirect treatment comparison between ribociclib vs. palbociclib (one-stage model: HR = 1.094, 95%-CI: 0.825–1.451, *p* = 0.531; two-stage model: HR = 0.858, 95%-CI: 0.594–1.239, *p* = 0.415), abemaciclib vs. palbociclib (one-stage model: HR = 0.790, 95%-CI: 0.583–1.071, *p* = 0.129; two-stage model: HR = 0.790, 95%-CI: 0.514–1.216, *p* = 0.285) and abemaciclib vs. ribociclib (one-stage model: HR = 0.722, 95%-CI: 0.520–1.002, *p* = 0.051; two-stage model: HR = 0.921, 95%-CI: 0.597–1.420, *p* = 0.710) failed to demonstrate a significant PFS difference (Figure 3A and Figure 4). The proportional hazards assumption was not violated (Appendix A). Comparisons against placebo arm within the one-stage indirect treatment comparison network yielded similar results (Appendix A). 

### 3.5. Overall Survival

Two phase III randomized controlled trials, PALOMA-2 [7], MONALEESA-2 [5] and a pre-specified interim OS analysis of MONARCH-3 [6] comprising 1827 patients were included in this analysis. Likewise, indirect treatment comparison between ribociclib vs. palbociclib (one-stage model: HR = 0.903, 95%-CI: 0.746–1.094, *p* = 0.297; two-stage model: HR = 0.821, 95%-CI: 0.614–1.098, *p* = 0.183), abemaciclib vs. palbociclib (one-stage model: HR = 0.843, 95%-CI: 0.690–1.030, *p* = 0.094; two-stage model: HR = 0.815, 95%-CI: 0.585–1.135, *p* = 0.227) and abemaciclib vs. ribociclib (one-stage model: HR = 0.933, 95%-CI: 0.753–1.157, *p* = 0.528; two-stage model: HR = 0.993, 95%-CI: 0.719–1.372, *p* = 0.966) failed to demonstrate a significant OS difference (Figure 3B and Figure 4). The proportional hazards assumption was not violated (Appendix A). Comparisons against placebo arm within the one-stage indirect treatment comparison network yielded similar results (Appendix A). 

The risk-of-bias assessment demonstrated low risk across all 3 trials (Table 2). GRADE assessment demonstrates high level of certainty of the computed results (Appendix A). Publication bias and statistical heterogeneity was not evaluable as each CDK4/6-inhibitor was only represented by one trial.

## 4. Discussion

Our analyses based on patient-level indirect treatment comparison between CDK4/6-inhibitors in post-menopausal patients with HR+/HER2− MBC showed no significant difference in OS and PFS, as shown through the Cox proportion hazard model with 2-stage sensitivity analyses. Furthermore, reconstruction of KM curves through graphical reconstruction showed non-significant differences in both the PFS and OS curves between the 3 study agents.

In HR+/HER2− metastatic breast cancer, endocrine therapy is preferred over chemotherapy in patients without immediate visceral crisis given its ability to achieve comparable survival outcomes with a more favorable toxicity profile and quality of life [17]. Preclinical work has demonstrated that tumor cell proliferation in breast cancer is significantly driven by upregulation of the cyclin D–CDK4/6-axis [18,19], and CDK4/6-inhibitors induce cytostasis through cell-cycle arrest in the G1 phase leading to in-growth inhibition. making inhibition of this pathway an attractive strategy to overcome resistance seen with endocrine therapy [20].

Over the past decade, the combination of CDK4/6-inhibitor and endocrine therapy has been established as standard-of-care for first-line treatment of post-menopausal patients with HR+/HER2− MBC. All 3 United States Food and Drug Administration (FDA) approved agents largely perceived to have similar clinical efficacy in terms of improvement in response rates and PFS when added to endocrine therapy. Nonetheless, differences between the three CDK4/6-inhibitors are present from a molecular standpoint. Both palbociclib and ribociclib are derived from a similar pyrido[2,3-d]-pyrimidin-7-one scaffold that was optimized for selectivity toward CDK4/6 [21], and pre-clinical benchwork has shown similar potencies in terms of CDK4 and CDK6 inhibition [22]. Conversely, abemaciclib has been shown to have five-fold more potency for CDK4 and inhibit multiple other CDK kinase activities [23]. Unfortunately beyond the bench, there remains a dearth of head-to-head clinical comparisons between CDK4/6-inhibitors to address this dilemma.

With all three first-line studies in post-menopausal women showing similar HR for PFS, physicians have generally deemed the three CDK4/6 inhibitors as equivalent from the efficacy standpoint, and treatment choice is largely guided by patient comorbidity, toxicity profile, and cost factors. However, the lack of OS benefit in PALOMA-2 challenges the notion that efficacy of CDK4/6-inhibitors may be presumed to be the same, and whether palbociclib may be inferior as a treatment choice compared to ribociclib and abemaciclib.

In this patient-level indirect treatment comparison of CDK4/6-inhibitors amongst post-menopausal patients with HR+/HER2− MBC, no significant difference in OS and PFS was found. The lack of significant differences in PFS is consistent with prior studies that similar efficacy and overall activity among the three CDK4/6 inhibitors [24], such as a prior meta-analysis that has shown superiority of CDK4/6 inhibitor plus endocrine therapy in the first and second line setting in terms of PFS compared to highly active anthracycline or taxane-based chemotherapy, with no reported significant differences between the three CDK4/6 inhibitors [24]. Our results are the first to report a comparison of OS outcomes across the 3 CDK4/6 inhibitor agents, and showed no significant OS differences amongst these agents. While head-to-head studies are the gold-standard for comparison across agents, the lack of these studies negates the possibility of such comparison, and our analyses help provide insight in terms of comparative survival efficacy. Our results are further supported by real-word study from the Flatiron database demonstrating an overall survival benefit with the addition of palbociclib to endocrine therapy compared to endocrine therapy alone (propensity-score matched analysis: palbociclib + aromatase inhibitor median overall survival [mOS] 57.8 months vs. aromatase inhibitor alone mOS 43.5 months; HR = 0.72, 95%-CI: 0.62–0.83, *p* < 0.0001) [25]. While there may still be preference for ribociclib/abemaciclib over palbociclib based on the HR from individual studies, the lack of significant survival difference between the 3 FDA approved agents supports the use of palbociclib as first line therapy in HR+/HER2− MBC, especially in patients who have already been commenced on and derive ongoing clinical benefit from this agent. Additionally, in patients with comorbidities that may predispose them to cardiovascular or hepatic adverse events that are more prominent with the other agents, palbociclib remains a reasonable treatment option. In the same vein, a commentary by Grinshpun et al. also recommended that patients already on a palbociclib-based regimen without intolerable toxicity should continue with this active regimen instead of switching to an alternative CDK4/6-inhibitor [26].

Several limitations of this study are acknowledged. First, the reconstruction algorithm does not enable retrieval of patient-level covariates, rendering it not possible to adjust for confounders. Further analyses with individual patient data may allow for better adjustment of confounders to provide better granularity for the comparision of survival outcomes across different agents, but this data is not readily available in the public domain, thus limiting its feasibility. Second, competing risks could not be accounted for in the analysis of PFS. This is because time-to-event data was derived per plot, multiple survival endpoints could not be attained for each patient row in the retrieved dataset. Third, slightly differing endocrine backbones were utilized. Patients in PALOMA-2 and MONALEESA-2 were administered letrozole while MONARCH-3 utilized either anastrozole or letrozole as endocrine therapy of choice. Nonetheless, majority of patients (79.1%) in MONARCH-3 were treated with letrozole, and efficacy of letrozole and anastrazole, both non-steroidal aromatase inhibitors are generally viewed as equivalent, thus this difference is believed to be minimal on treatment effect. Fourth, cross-trial baseline imbalances may have compromised the transitivity assumption in the indirect treatment comparison network. As alluded earlier, a greater proportion of patients in PALOMA-2 had DFI <12 months, and a greater proportion of patients in MONARCH-3 had newly diagnosed metastatic disease. Patients with short DFI have poorer prognosis, while patients with de novo metastatic disease have better survival compared to patients with relapsed disease [27,28], thus possibly accounting for differences in overall survival that may be observed across studies. Finally, PALOMA-2 had a higher percentage of patients missing survival data than MONALEESA-2, and combined with an unknown rate of complete crossover, [26] this may have led to some bias in the comparative IPD analysis. Looking forward, identification of biomarkers that may help predict for response is paramount to allow for optimal patient selection between agents. In the same vein, understanding of mechanisms of tumor resistance to CDK4/6-inhibitor therapy will help guide choice for further treatment lines.

## 5. Conclusions

This meta-analysis performed indirect treatment comparisons and yielded comparable PFS and OS between three CDK4/6-inhibitors. As we await final OS data from MONARCH-3 and head-to-head comparison data of ribociclib and palbociclib from the ongoing HARMONIA trial (NCT05207709), our findings support all three drugs as options in first-line treatment in combination with endocrine therapy for post-menopausal patients with metastatic HR+ HER2− MBC.

## Figures and Tables

**Figure 1 cancers-15-04558-f001:**
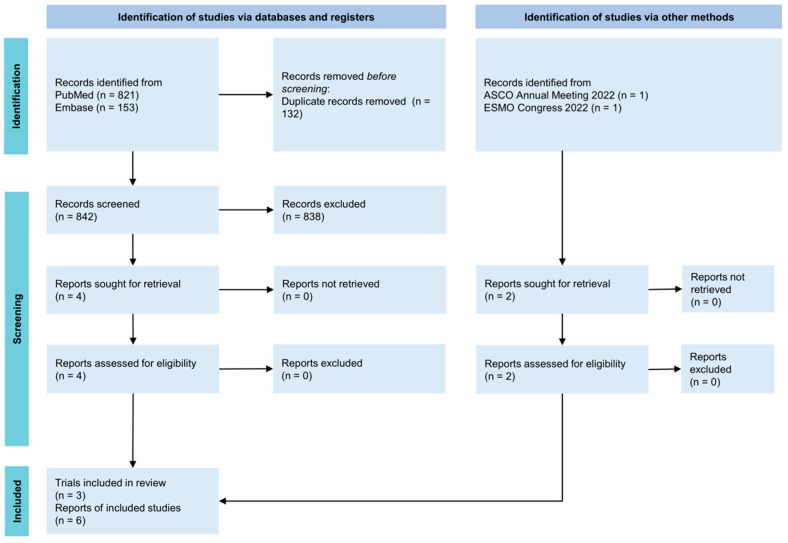
PRISMA diagram. PRISMA, Preferred Reporting Items for Systematic Reviews and Meta-Analyses; ASCO, American Society of Clinical Oncology Annual Meeting; ESMO, European Society for Medical Oncology Congress.

**Figure 2 cancers-15-04558-f002:**
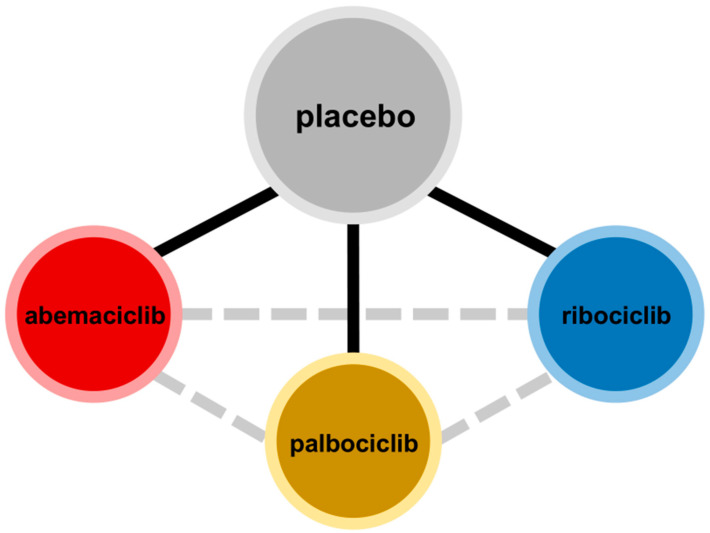
Network diagram. Black solid lines denote direct comparisons, grey dashed lines denote indirect comparisons. Node sizes are proportionate to the number of patients in the arm.

**Figure 3 cancers-15-04558-f003:**
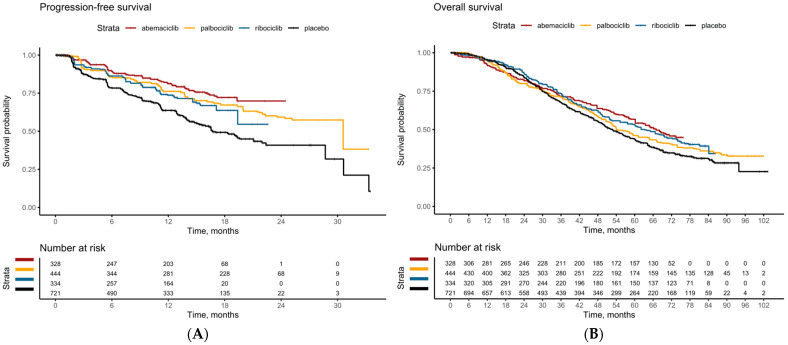
Pooled survival curves for (**A**) Progression-free survival (**B**) Overall survival of first-line CDK4/6 inhibitors in post-menopausal patients with HR+/HER2− metastatic breast cancer. CDK4/6-inhibitors, cyclin-D–cyclin-dependent kinase 4/6-inhibitors; HR+, hormone receptor-positive; HER2−, human epidermal growth factor receptor 2 negative.

**Figure 4 cancers-15-04558-f004:**
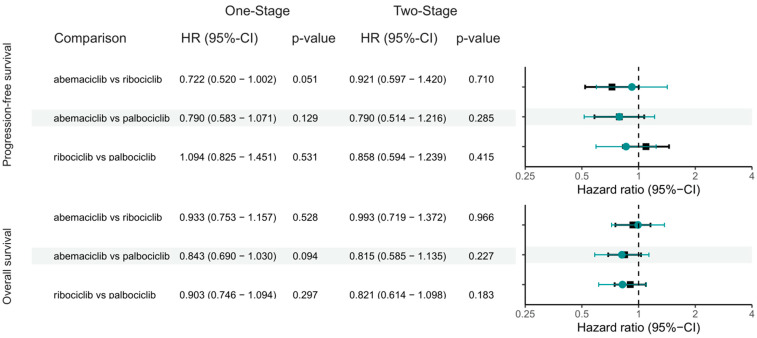
Interval plots with direct and indirect treatment comparison outcomes. HR, hazard ratio; CI, confidence interval; Hazard ratios and confidence intervals in black and turquoise correspond to results from the one-stage and two-stage analysis respectively.

**Table 1 cancers-15-04558-t001:** Baseline characteristics of included studies.

Study	Arm	Number of Patients	Age ^a^	ECOG PS (%)			PR+ (%)	Newly Diagnosed Metastatic Disease (%)	Prior Chemotherapy (%)	Prior Endocrine Therapy (%)	DFI for Existing Disease (%)	Number of Involved Organ Sites (%)			Visceral Disease (%)
				0	1	2						1	2	≥3	
MONALEESA-2 NCT01958021	Ribociclib	334	62 (23–91)	61.1	38.9	-	81.1	34.1	43.7	52.4	≤12 mo: 1.2 12–24 mo: 4.2 >24 mo: 60.5	29.9	35.3	34.1	59.0
	Placebo	334	63 (29–88)	60.5	39.5	-	83.2	33.8	43.4	51.2	≤12 mo: 3.0 12–24 mo: 4.5 >24 mo: 58.4	35.0	30.8	33.8	58.7
MONARCH-3 NCT02246621	Abemaciclib	328	63 (38–87)	58.5	41.5	-	77.7	41.2	38.1	45.7	<12 mo: excluded <36 mo: 28.0 ≥36 mo: 62.7	29.3	23.2	47.0	52.4
	Placebo	165	63 (32–88)	63.0	37.0	-	77.0	37.0	40.0	48.5	<12 mo: excluded <36 mo: 40.0 ≥36 mo: 50.0	28.5	25.5	45.5	53.9
PALOMA-2 NCT01740427	Palbociclib	444	62 (30–89)	57.9	40.1	2.0	NR	31.3	48.0	56.1	≤12 mo: 22.3 >12 mo: 40.1	31.1	26.4	42.5	48.2
	Placebo	222	61 (28–88)	45.9	52.7	1.4	NR	32.0	49.1	56.8	≤12 mo: 21.6 >12 mo: 41.9	29.7	23.4	46.9	49.5

^a^ Expressed as median (range). NR, not reported; HR, hormone receptor; HER2, human epidermal growth factor receptor 2; RT, radiotherapy; ECOG, Eastern Cooperative Oncology Group; PS, performance status; RECIST, Response Evaluation Criteria in Solid Tumours; PR, progesterone receptor.

**Table 2 cancers-15-04558-t002:** Cochrane Risk-of-Bias Tool evaluation of included studies.

Study	Outcomes	Randomization Process	Deviations from Intended Interventions	Missing Outcome Data	Measurement of the Outcome	Selection of the Reported Result	Overall
MONALEESA-2 NCT01958021	OS, PFS	⨁ Low	⨁ Low	⨁ Low	⨁ Low	⨁ Low	⨁ Low
MONARCH-3 NCT02246621	OS, PFS	⨁ Low	⨁ Low	⨁ Low	⨁ Low	⨁ Low	⨁ Low
PALOMA-2 NCT01740427	OS, PFS	⨁ Low	⨁ Low	⨁ Low	⨁ Low	⨁ Low	⨁ Low

OS, overall survival; PFS, progression-free survival.

## Data Availability

Data in this study were retrieved from previously published articles. There are no new data associated with this article.

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
