# Peer review of "Indirect Treatment Comparison of First-Line CDK4/6-Inhibitors in Post-Menopausal Patients with HR+/HER2− Metastatic Breast Cancer"

_cancers, 2023, doi:10.3390/cancers15184558_

Round 1

Reviewer 1 Report

This is a well written re-analysis of available trial data accross several trials. In the real-world setting this is consistent with what we see, and reassuring from that point of view.

Cross-trial comparisons are however always imperfect. The analysis methods used in this paper are an improvement on "eye-balling the curves" but remains imperfect. 

Author Response

This is a well written re-analysis of available trial data across several trials. In the real-world setting this is consistent with what we see, and reassuring from that point of view.

Cross-trial comparisons are however always imperfect. The analysis methods used in this paper are an improvement on "eye-balling the curves" but remains imperfect. 

Response: Thank you for your comments. We agree with the Reviewer that cross-trial comparisons are not ideal and randomised control studies remain the gold standard. However, with the lack of head-to-head comparisons, our study aims to use a novel statistical methodology to allow for comparisons as a surrogate for the lack of head-to-head clinical trials comparing the CDK4/6 inhibitors. Our methodology harness graphical plot digitization and computational inference to derive individual patient data directly from graphs and figures presented in trial manuscripts, and help provide insight in terms of comparative survival efficacy.

Nonetheless, this remains imperfect, and we have acknowledged the statistical limitations of our methodology in the Discussion:

“Second, competing risks could not be accounted for in the analysis of PFS. This is because time-to-event data was derived per plot, multiple survival endpoints could not be attained for each patient row in the retrieved dataset

...

Fourth, cross-trial baseline imbalances may have compromised the transitivity assumption in the indirect treatment comparison network”

Reviewer 2 Report

1 “This study sought to elucidate indirect survival outcomes between CDK4/6-inhibitors in this setting", needs more detail.

2 The main contributions of the manuscript are not clear. The main contributions of the ‎article must be very clear and would be better if summarize ‎them into 3-4 points at the ‎end of the introduction.‎

3 “A search was conducted from inception to 15th September 2022”, from start to end should be some graphs.

4 The introduction section needs to be improved. An introduction is an important road map for the rest of the paper that should be consist of an opening hook to catch the researcher's attention, relevant background study, and a concrete statement that presents main argument but your introduction lacks these fundamentals, especially relevant background studies. This related work is just listed out without comparing the relationship between this paper's model and them; only the method flow is introduced at the end; and the principle of the method is not explained. To make soundness of your study must include these latest related works.

I (2022) Calcium Homeostasis in Parkinson’s Disease: From Pathology to Treatment. Neurosci. Bull. 38, 1267–1270. https://doi.org/10.1007/s12264-022-00899-6

II (2023). Royal jelly acid suppresses hepatocellular carcinoma tumorigenicity by inhibiting H3 histone lactylation at H3K9la and H3K14la sites. Phytomedicine, 154940. doi: https://doi.org/10.1016/j.phymed.2023.154940

III (2022). Transcranial alternating current stimulation for treating depression: a randomized controlled trial. Brain, 145(1), 83-91. doi: 10.1093/brain/awab252

IV (2022). The Effect of Sevoflurane on the Proliferation, Epithelial-Mesenchymal Transition (EMT) and Apoptosis in Human Breast Cancer Cells. Journal of Biological Regulators and Homeostatic Agents, 36(3), 583-592. doi: 10.23812/j.biol.regul.homeost.agents.20223603.66

V (2023). Identifying Malignant Breast Ultrasound Images Using ViT-Patch. Applied Sciences, 13(6), 3489. doi: 10.3390/app13063489"

5 The authors need to add new figures to show the main structure of the proposed system. ‎This will help the reader to get a better understanding of what is going on in the proposed ‎system.‎

6 There is much literature on this topic already; why did it need this work?

7 “KM plots were extracted from original reports and pre-processed”, where is that?

8 Mention the limitations and future works of the developed system elaborately.

Minor editing of English language required.

Author Response

1 “This study sought to elucidate indirect survival outcomes between CDK4/6-inhibitors in this setting", needs more detail.

Response: We have modified the sentence in the Abstract to “With the lack of head-to-head comparison studies, our study sought to compare indirect survival outcomes between CDK4/6-inhibitors in this setting using a novel reconstructive algorithm”. Further description of the this is elaborated in the methods section of the abstract.

2 The main contributions of the manuscript are not clear. The main contributions of the ‎article must be very clear and would be better if summarize ‎them into 3-4 points at the ‎end of the introduction.‎

Response: Our main contributions are a suggestion that there is no significant overall or progression-free survival differences between all CDK4/6-inhibitor agents plus an aromatase inhibitor in post-menopausal patients with HR+/HER- MBC. We believe these findings would have an immediate effect on clinical practice, as it provides oncologists reassurance to keep patients on their current CDK4/6 inhibitor regimens, without the need to discuss changing them urgently based on the recent data presented.

We have modified our manuscript in the Introduction section to set the stage and aims for our manuscript, including how we believe our methodology and results can contribute to the field.

A discussion of the contributions based on the results of the manuscript are detailed in the Discussion and Conclusion.

3 “A search was conducted from inception to 15th September 2022”, from start to end should be some graphs.

Response: We refer to Figure 1 in our manuscript, which is the PRISMA flow diagram illustrating the flow of our statistical analysis from database search to screening to study inclusion. Moreover, our search strings for respective databases are detailed in Supplemental Table 1. These conform to the PRISMA guidelines for standardized reporting in meta-analysis.

4 The introduction section needs to be improved. An introduction is an important road map for the rest of the paper that should be consist of an opening hook to catch the researcher's attention, relevant background study, and a concrete statement that presents main argument but your introduction lacks these fundamentals, especially relevant background studies. This related work is just listed out without comparing the relationship between this paper's model and them; only the method flow is introduced at the end; and the principle of the method is not explained. To make soundness of your study must include these latest related works.

I (2022) Calcium Homeostasis in Parkinson’s Disease: From Pathology to Treatment. Neurosci. Bull. 38, 1267–1270. https://doi.org/10.1007/s12264-022-00899-6

II (2023). Royal jelly acid suppresses hepatocellular carcinoma tumorigenicity by inhibiting H3 histone lactylation at H3K9la and H3K14la sites. Phytomedicine, 154940. doi: https://doi.org/10.1016/j.phymed.2023.154940

III (2022). Transcranial alternating current stimulation for treating depression: a randomized controlled trial. Brain, 145(1), 83-91. doi: 10.1093/brain/awab252

IV (2022). The Effect of Sevoflurane on the Proliferation, Epithelial-Mesenchymal Transition (EMT) and Apoptosis in Human Breast Cancer Cells. Journal of Biological Regulators and Homeostatic Agents, 36(3), 583-592. doi: 10.23812/j.biol.regul.homeost.agents.20223603.66

V (2023). Identifying Malignant Breast Ultrasound Images Using ViT-Patch. Applied Sciences, 13(6), 3489. doi: 10.3390/app13063489"

Response: We thank the Reviewer for these comments. Our introduction has put forth the roadmap of the paper and set the stage by listing the three landmark RCTs (PALOMA-2, MONALEESA-2 and MONARCH-3) and stating the current controversy regarding inconsistency in overall survival. The last paragraph of our introduction has been reworded to signpost our paper’s workflow:

“With the lack of head-to-head RCTs comparing the three agents, we sought to compare survival outcomes between three different CDK4/6-inhibitor agents to provide additional evidence towards treatment choices. Advances in graphical plot digitization and computational inference now allows for derivation of individual patient data directly from graphs and figures presented in manuscripts and conference abstracts. This technique is now gaining traction allowing for large individual patient-data secondary analyses to be performed. Hence, we aimed to harness this technique to conduct a patient-level indirect treatment comparison of OS and PFS between first-line CDK4/6-inhibitor agents in HR+/HER2- metastatic breast cancer.”

The five latest related works suggested by the Reviewer covers a wide range of subjects, we respectfully disagree should be included as refences for our manuscript. Suggested references 1 and 3 are in the field of neuroscience, and reference 2 and 4 discusses preclinical studies on tumorigenicity unrelated to our current study of CDK4/6-inhibitor agents in HR+/HER2- metastatic breast cancer. For reference V, despite its relation to breast malignancies, this is an imaging study which is not related to our study of treatment strategy for metastatic breast cancer.

5 The authors need to add new figures to show the main structure of the proposed system. ‎This will help the reader to get a better understanding of what is going on in the proposed ‎system.‎

Response: Thank you for this comment. As mentioned above in our response to point 3, we followed the PRISMA guidelines for meta-analysis and have provided Figure 1 to show the flow of our study. Moreover, we have shown the comparisons of reconstructed and original Kaplan-Meier curves in Supplemental Table 2, which forms the basis for our statistical analysis wherein pooled curves for OS and PFS were generated. The statistical linchpin of this analysis is the reconstruction of Kaplan-Meier curves, which we have described briefly in our Methods (see response to point 7 below); this has been previously well-described in literature and our group has published several analyses using these methods, e.g.

  • Syn NL, Cummings DE, Wang LZ, et al. Association of metabolic–bariatric surgery with long-term survival in adults with and without diabetes: a one-stage meta-analysis of matched cohort and prospective controlled studies with 174 772 participants. The Lancet. 2021.
  • Tan BKJ, Han R, Zhao JJ, et al. Prognosis and persistence of smell and taste dysfunction in patients with covid-19: meta-analysis with parametric cure modelling of recovery curves. BMJ. 2022;378:e069503.
  • Zhao JJ, Yap DWT, Chan YH, et al. Low Programmed Death-Ligand 1–Expressing Subgroup Outcomes of First-Line Immune Checkpoint Inhibitors in Gastric or Esophageal Adenocarcinoma. Journal of Clinical Oncology. 2021:JCO.21.01862.
  • Fong KY, Zhao JJ, Sultana R, et al. First-Line Systemic Therapies for Advanced Hepatocellular Carcinoma: A Systematic Review and Patient-Level Network Meta-Analysis. Liver Cancer. 2022 Aug 23;12(1):7-18.

6 There is much literature on this topic already; why did it need this work?

Response: Thank you for this comment. We refer to our response to Point 4. Briefly, the three FDA-approved CDK4/6-inhibitors are assumed to have similar clinical efficacy based on previously published progression-free survival data. However, recently disparate readout for overall survival of these CDK4/6-inhibitor agents has challenged this notion. Hence, the unmet clinical need of our study is the comparison of survival outcomes across the different CDK4/6-inhibitor agents (palbociclib, ribociclib and abemaciclib). Given that there are no head-to-head trials of agents versus each other, the best way to compare these is via patient-level indirect analysis as we have done.

7 “KM plots were extracted from original reports and pre-processed”, where is that?

Response: We apologize for the confusion that this sentence may have caused. We have modified this section of the manuscript to add further clarity to the statistical methods:

“Briefly, images of KM plots were extracted from original reports and digitized. The graphical reconstructive algorithm was employed to reconstruct patient-level survival data from digitized KM curves by back-solving the KM equations, utilizing data from the risk table. Hazard ratios (HRs) were computed from these individual patient data using a marginal Cox proportional hazards model.”

8 Mention the limitations and future works of the developed system elaborately.

Response: We thank the Reviewer for his comments and suggestions. In our revision, we have expanded on the discussion and also written a separate section on conclusions, stating the limitations in the second last paragraph of the manuscript. We hope that these sections of our manuscript will give readers an adequate view of the future directions of research in this domain while keeping in mind the limitations of our present analysis.

Reviewer 3 Report

The authors aimed to elucidate indirect survival outcomes between CDK4/6-inhibitors, as an effective first-line endocrine treatment of patients with hormone-receptor positive (HR+)/human-epidermal-growth-factor-receptor-2 negative (HER2-) metastatic breast cancer (MBC) in the endpoint of progression-free survival (PFS). A

Their findings suggest that the efficacy of the 3 CDK4/6 inhibitors is comparable, supporting all 3 drugs as options in first-line treatment in combination with endocrine therapy for post-menopausal patients with metastatic HR+ HER2- MBC.

While the results and conclusions are not novel, they are based on solid data and help to extend the knowledge on the topic, which may be beneficial in the future.

Therefore, I have no further comments against the manuscript.

Author Response

The authors aimed to elucidate indirect survival outcomes between CDK4/6-inhibitors, as an effective first-line endocrine treatment of patients with hormone-receptor positive (HR+)/human-epidermal-growth-factor-receptor-2 negative (HER2-) metastatic breast cancer (MBC) in the endpoint of progression-free survival (PFS). A

Their findings suggest that the efficacy of the 3 CDK4/6 inhibitors is comparable, supporting all 3 drugs as options in first-line treatment in combination with endocrine therapy for post-menopausal patients with metastatic HR+ HER2- MBC.

While the results and conclusions are not novel, they are based on solid data and help to extend the knowledge on the topic, which may be beneficial in the future.

Therefore, I have no further comments against the manuscript.

Response: We are grateful to the Reviewer for their kind comments and this is in line with our hope that our paper will further solidify the data showing comparability of the three CDK/6 inhibitors.

Reviewer 4 Report

The abstract needs modification. The introduction part may be improved with additional references. Need for this research has to be included. Section 2 needs clarification and methodology tree diagram. All the tables must be enhanced with good quality. Section 3.5 needs some statistical quantification. The discussion section is too lengthy may be reduced with key points. Limitations are good. Conclusion/Summary may be included. References are adequate. The methods and experimental procedure are at par with standard one.

NIL

Author Response

The abstract needs modification.

Response: We have modified the abstract as suggested. We note that the abstract is meant to convey the salient points of our manuscript while being bounded by a word limit and hope that it is adequate to convey the message of our manuscript.

The introduction part may be improved with additional references. Need for this research has to be included.

Response: We thank the Reviewer for these comments. Our introduction has put forth the roadmap of the paper and set the stage by listing the three landmark RCTs (PALOMA-2, MONALEESA-2 and MONARCH-3) and stating the current controversy regarding inconsistency in overall survival. The last paragraph of our introduction has been reworded to signpost our paper’s workflow:

“With the lack of head-to-head RCTs comparing the three agents, we sought to compare survival outcomes between three different CDK4/6-inhibitor agents to provide additional evidence towards treatment choices. Advances in graphical plot digitization and computational inference now allows for derivation of individual patient data directly from graphs and figures presented in manuscripts and conference abstracts. This technique is now gaining traction allowing for large individual patient-data secondary analyses to be performed. Hence, we aimed to harness this technique to conduct a patient-level indirect treatment comparison of OS and PFS between first-line CDK4/6-inhibitor agents in HR+/HER2- metastatic breast cancer.”

Regarding the need for this research, the three FDA-approved CDK4/6-inhibitors are assumed to have similar clinical efficacy based on previously published progression-free survival data. However, recently disparate readout for overall survival of these CDK4/6-inhibitor agents has challenged this notion. Hence, the unmet clinical need of our study is the comparison of survival outcomes across the different CDK4/6-inhibitor agents (palbociclib, ribociclib and abemaciclib). Given that there are no head-to-head trials of agents versus each other, the best way to compare these is via patient-level indirect analysis as we have done.

Section 2 needs clarification and methodology tree diagram.

Response: We have added in further details on the Kaplan-Meier curve analysis. For the methodology tree diagram, we refer to Figure 1 in our manuscript, which is the PRISMA flow diagram illustrating the flow of our statistical analysis from database search to screening to study inclusion. These conform to the PRISMA guidelines for standardized reporting in meta-analysis.

All the tables must be enhanced with good quality.

Response: Thank you for this comment. We have provided the tables in vector-based PDF form since submission; we hope that this issue will be rectified during publication.

Section 3.5 needs some statistical quantification.  

Response: In section 3.5 (Results; Overall survival), we have already quantified our indirect analysis using hazard ratios, 95% confidence intervals, and p-values. Figures and Supplementary Figures are also provided.

The discussion section is too lengthy may be reduced with key points. Limitations are good. Conclusion/Summary may be included.  

Response: We have divided out and modified the last paragraph as our conclusion and reorganized the discussion slightly to state key points. We note that our paper was originally flagged by the journal editors for being shorter than usual recommended manuscript length, hence we are keen to keep the rest of the Discussion section given that it states our main points as well as its relation to recent, relevant literature.

References are adequate. The methods and experimental procedure are at par with standard one.

Response: Thank you for these kind comments.

Round 2

Reviewer 2 Report

Accept

Reviewer 4 Report

All the corrections are included in the paper. Hence, there is no need for further review.

NIL